# Comparison of Four Commercially Available Point-of-Care Tests to Detect Antibodies against Canine Parvovirus in Dogs

**DOI:** 10.3390/v13010018

**Published:** 2020-12-23

**Authors:** Michèle Bergmann, Mike Holzheu, Yury Zablotski, Stephanie Speck, Uwe Truyen, Reinhard K. Straubinger, Katrin Hartmann

**Affiliations:** 1Clinic of Small Animal Medicine, Centre for Clinical Veterinary Medicine, LMU Munich, Veterinaerstrasse 13, 80539 Munich, Germany; Y.Zablotski@med.vetmed.uni-muenchen.de (Y.Z.); hartmann@lmu.de (K.H.); 2Clinic of Small Animal Surgery and Reproduction, Centre for Clinical Veterinary Medicine, LMU Munich, Veterinaerstrasse 13, 80539 Munich, Germany; M.Holzheu@medizinische-kleintierklinik.de; 3Institute of Animal Hygiene and Veterinary Public Health, University of Leipzig, An den Tierkliniken 1, 04103 Leipzig, Germany; stephanie@speck-kaysser.de (S.S.); truyen@vetmed.uni-leipzig.de (U.T.); 4Bacteriology und Mycology, Institute for Infectious Diseases and Zoonoses, Faculty of Veterinary Medicine, LMU Munich, Veterinaerstrasse 13, 80539 Munich, Germany; Reinhard.Straubinger@micro.vetmed.uni-muenchen.de

**Keywords:** POC test, in-house test, CPV, protection, vaccination, sensitivity, specificity, PPV, NP

## Abstract

Measuring antibodies to evaluate dogs’ immunity against canine parvovirus (CPV) is useful to avoid unnecessary re-vaccinations. The study aimed to evaluate the quality and practicability of four point-of-care (POC) tests for detection of anti-CPV antibodies. The sera of 198 client-owned and 43 specific pathogen-free (SPF) dogs were included; virus neutralization was the reference method. Specificity, sensitivity, positive and negative predictive value (PPV and NPV), and overall accuracy (OA) were calculated. Specificity was considered to be the most important indicator for POC test performance. Differences between specificity and sensitivity of POC tests in the sera of all dogs were determined by McNemar, agreement by Cohen’s kappa. Prevalence of anti-CPV antibodies in all dogs was 80% (192/241); in the subgroup of client-owned dogs, it was 97% (192/198); and in the subgroup of SPF dogs, it was 0% (0/43). FASTest^®^ and CanTiCheck^®^ were easiest to perform. Specificity was highest in the CanTiCheck^®^ (overall dogs, 98%; client-owned dogs, 83%; SPF dogs, 100%) and the TiterCHEK^®^ (overall dogs, 96%; client-owned dogs, 67%; SPF dogs, 100%); no significant differences in specificity were observed between the ImmunoComb^®^, the TiterCHEK^®^, and the CanTiCheck^®^. Sensitivity was highest in the FASTest^®^ (overall dogs, 95%; client-owned dogs, 95%) and the CanTiCheck^®^ (overall dogs, 80%; client-owned dogs, 80%); sensitivity of the FASTest^®^ was significantly higher compared to the one of the other three tests (McNemars *p*-value in each comparison: <0.001). CanTiCheck^®^ would be the POC test of choice when considering specificity and practicability. However, differences in the number of false positive results between CanTiCheck^®^, TiterCHEK^®^, and ImmunoComb^®^ were minimal.

## 1. Introduction

Canine parvovirus-2 (CPV-2) is a highly pathogenic virus, and protection is essential for all dogs [1,2]. Presence of antibodies against CPV in adult dogs suggests adequate immunity against disease [3,4]. Measurement of pre-vaccination antibodies against CPV is therefore useful to determine the specific immune status of an adult dog. If antibodies are present, unnecessary re-vaccinations, which potentially cause vaccine-associated adverse events, can be avoided [2,5]. Additionally, the measurement of antibodies can be used to determine whether dogs responded to vaccination, as well as for the management of disease outbreaks, e.g., in animal shelters [2,6,7].

Virus neutralization (VN) detects antibodies that neutralize infectious particles and prevent infection [8]. VN can be used to determine the level of anti-CPV antibody titers, but it can only be performed in specialized laboratories [8]. Thus, point-of-care (POC) tests would be a useful tool in practice, to assess whether a dog is potentially unprotected and requires re-vaccination.

Recently, four different POC tests became available on the market, for detection of anti-CPV antibodies in-house in veterinary practice (Table 1). It takes between 10 and 21 min to perform these tests.

The TiterCHEK^®^ CDV/CPV (Zoetis) has been evaluated previously by three studies [9,10,11], and one of these studies compared its quality to that of the ImmunoComb^®^ Canine VacciCheck (Biogal Laboratories) [10]. The FASTest^®^ CPV-CDV (MEGACOR Diagnostik GmbH) has been evaluated by only one study [12]. So far, there are no data (other than the manufacturers´ information) on the CanTiCheck^®^ (Fassisi Gesellschaft für Veterinärdiagnostik und Umweltanalysen mbH).

The aim of this study was to compare the four commercially available POC tests detecting antibodies against CPV, concerning their practicability, specificity, sensitivity, positive predictive value (PPV), negative predictive value (NPV), and overall accuracy (OA), using VN as the reference standard.

## 2. Materials and Methods

### 2.1. Sera

The protocol of this prospective study was accepted by the ethical committee of the LMU Munich, Germany (reference number 124-13-05-2018) on 18 July 2018.

Client-owned dogs (*n* = 198) that were presented to the Clinic of Small Animal Medicine, LMU, Munich, from June to August 2018, and that needed blood analyses for healthcare assessment or diagnostic purposes were included. The dogs´ age ranged from 3 months to 16 years (median age was 9 years). Table 2 shows the dogs´ signalment and health status.

In addition, 43 samples of specific pathogen-free (SPF) dogs that had never been vaccinated were included in the study. These stored samples were provided by the Institute of Animal Hygiene and Veterinary Public Health, University of Leipzig, and the Institute for Infectious Diseases and Zoonoses, LMU Munich. All samples were stored at −80 °C, until testing.

### 2.2. Measurement of Anti-CPV Antibodies by VN

Crandell Rees feline kidney (CRFK) cells were maintained in Dulbecco’s MEM (Merck Millipore, Darmstadt, Germany) supplemented with 5% fetal calf serum (FCS; Merck Millipore), 1% nonessential amino acids (Merck Millipore), and 1% penicillin–streptomycin (Merck Millipore), at 37 °C, with 5% CO_2_. Serum samples underwent heat-inactivation (56 °C for 30 min). A 100 µL aliquot of each serum was pre-diluted (1:5) in phosphate-buffered saline (PBS; pH 7.2) and further serially diluted at steps of 1:2. Each dilution was mixed with an equal volume of CPV-2 strain vBI 265 (200 median tissue culture infective dose per 0.1 mL) and incubated at 37 °C for 60 min. Subsequently, CRFK cells seeded in 96-well microtiter plates were inoculated with 100 μL of these serum/virus mixtures. Plates were incubated for five days, at 37 °C, with 5% CO_2_. Thereafter, culture supernatants were taken off, and cells were fixed by using acetone (>99·9%)/methanol (>99·9%) 1:1 (vol/vol) at −20 °C for 20 min. Subsequently, the fixation medium was discarded, and cells were allowed to dry for 30 min. After a single PBS-washing step, blocking was performed in PBS, with 3% FCS, to reduce non-specific binding and background signal. For virus staining, an anti-CPV monoclonal antibody [13] was added, and samples were incubated at 37 °C for 6–8 hours. After washing with PBS, a fluorescein isothiocyanate-conjugated goat-anti-mouse IgG (H + L) conjugate (Dianova, Hamburg, Germany) was added (room temperature, overnight). Samples were analyzed by using a Leica DMIL fluorescence microscope. The positive control serum was obtained from a privately owned vaccinated dog. All samples were run in duplicate in the same test. A titer <10 was considered negative.

### 2.3. Measurement of Anti-CPV Antibodies by POC Tests

The four POC tests (ImmunoComb^®^ Canine Vaccicheck; TiterCHEK^®^ CDV/CPV; FASTest^®^ CPV-CDV, CanTiCheck^®^) measure antibodies against CPV, as well as against canine distemper virus (CDV) (TiterCHEK^®^; FASTest^®^, CanTiCheck^®^) or against CPV, CDV, and canine adenovirus (CAV) (ImmunoComb^®^); data on antibodies against CDV and CAV-1 were not the subject of the present study.

All tests were analyzed according to manufacturers´ instructions, by an independent person who was blinded to the results of the VN.

The ImmunoComb^®^ and the TiterCHEK^®^ are enzyme-linked immunosorbent assays; FASTest^®^ and CanTiCheck^®^ are lateral flow immunoassays (immunochromatography assays). Test results were regarded positive, as defined by the manufacturers´ instructions.

### 2.4. Statistical Analysis

The sample-size calculation for the sera of client-owned dogs was based on the assumption of a prevalence of 70% [5,14,15] and a 5% desired level of significance [16]. To achieve a power of ≥ 80% (based on results of Gray et al., 2012 [9] and Litster et al., 2012 [11]), at least 147 samples were required. Results of the four POC tests were compared to the results of the VN assay, for determination of the diagnostic quality of each assay. Sensitivity, specificity, PPV, NPV, and OA were calculated for all dogs, as well as for the subgroup of client-owned dogs and for the subgroup of SPF dogs. Specificity was regarded as the most important parameter for the quality of the tests. McNemar´s test was used for the paired study design, to determine significant differences in specificity and sensitivity between the POC tests for all dogs. To assess the agreement of the results among the four POC tests, Cohen´s Kappa statistic was used. The *p*-values < 0.05 were considered significant.

## 3. Results

### 3.1. Prevalence of Anti-CPV Antibodies Determined by VN

The prevalence of antibodies, when considering the sera of all dogs, was 80% (192/241; 95% CI: 74–85). Antibodies against CPV were present in 97% (192/198; 95% CI: 93–99) of the client-owned dog population. None of the SPF dogs had anti-CPV antibodies (0/43; 95% CI: 0–10).

### 3.2. Practicability of the POC Tests

Results of all serum samples in all of the four POC tests (TiterCHEK^®^, FASTest^®^, ImmunoComb^®^, and CanTiCheck^®^) could clearly be classified as positive or negative. According to the manufacturers´ instructions, the FASTest^®^ and the CanTiCheck^®^ were stored at room temperature. For testing, 5 and 50 µL of serum were necessary, respectively. In both tests, results were available after 10 min. The TiterCHEK^®^ and the ImmunoComb^®^ were refrigerated, and it was necessary to bring them to room temperature before measurement, according to the manufacturers´ recommendations; after the warm-up period, the tests delivered results after an additional 20 and 21 min, respectively. For testing, 1 and 5 µL of serum were necessary, respectively.

### 3.3. Measurement of Anti-CPV Antibodies by POC Tests

Sensitivity, specificity, PPV, NPV, and OA of the four POC tests, when compared to VN as the reference standard, in all groups of dogs are shown in Table 3 and Table 4. Specificity was highest in the CanTiCheck^®^ (overall dogs, 98%; client-owned dogs, 83%; SPF dogs, 100%) and the TiterCHEK^®^ (overall dogs, 96%; client-owned dogs, 67%; SPF dogs, 100%); no significant differences in specificity were observed between the ImmunoComb^®^, the TiterCHEK^®^, and the CanTiCheck^®^ when considering all dogs (Table 5). Sensitivity was highest in the FASTest^®^ (overall dogs, 95%; client-owned dogs, 95%) and the CanTiCheck^®^ (overall dogs, 80%; client-owned dogs, 80%); sensitivity of the FASTest^®^ was significantly higher compared to the one of the other three tests (McNemars *p*-value in each comparison: <0.001), sensitivity of the CanTiCheck^®^ was significantly higher compared to the one of the TiterCHEK^®^ (McNemars *p*-value: <0.001). Table 6 shows the agreement of the POC test results in all dogs. TiterCHEK^®^ and FASTest^®^ had the same result in 164/241 (68%) samples, TiterCHEK^®^ and ImmunoComb^®^ in 166/241 (69%), FASTest^®^ and ImmunoComb^®^ in 171/241 (71%), CanTiCheck^®^ and ImmunoComb^®^ in 186/241 (77%), CanTiCheck^®^ and TiterCHEK^®^ in 185/241 (77%), and CanTiCheck^®^ and FASTest^®^ in 196/241 (81%) samples.

Statistical analysis revealed a moderate agreement in the results of all dogs between the CanTiCheck^®^ and TiterCHEK^®^, κ = 0.53; between the CanTiCheck^®^ and the ImmunoComb^®^, κ = 0.52; and between CanTiCheck^®^ and the FASTest^®^, κ = 0.54. Agreement among the other POC tests was fair (Table 6). Table 7 shows the subgroups of VN (virus neutralization) test results and the respective results of the four POC (point-of-care) tests for the detection of antibodies against CPV (canine parvovirus) in all dogs.

## 4. Discussion

Although disease caused by CPV infection is rare in Northern and Central European countries today, the risk of spread of CPV via the import of dogs from Southern or Eastern Europe is still high [17,18]. Thus, all dogs should be protected against CPV infection at any time [2,8]. Detection of anti-CPV antibodies (at any level) is predictive for protection in adult dogs that have been vaccinated or infected previously [19]. Therefore, measurement of antibodies against CPV can be used to evaluate dogs´ individual immune status [2,8]. Reference tests to detect antibody titers, such as VN, have to be performed in specialized laboratories. This is inconvenient for healthcare appointments at the veterinarian, because the decision whether a dog should be vaccinated or not should ideally be made at the veterinarian´s practice, during the appointment. Thus, fast and reliable POC tests that can easily be performed in practice would be desirable. In Europe, four POC tests are available for measuring anti-CPV antibodies since recently. This is the first comparison study that evaluated and compared all four tests.

The FASTest^®^ and the CanTiCheck^®^ were the most practicable. Both tests can be stored at room temperature and do not need to be warmed up before testing, which makes the procedure less time-consuming and avoids incorrect handling. In addition, both tests were unproblematic to perform, as buffered-mixed serum can be easily applied to the test kit, and, thus, tests can be carried out by the veterinarian, without requiring any additional lab-trained staff.

Testing with the TiterCHEK^®^ and the ImmunoComb^®^ was more time-consuming compared to the FASTest^®^ and the CanTiCheck^®^, and the test procedures included several mixing steps, making these tests more laborious and less practicable.

For assessing performance of the pre-vaccination antibody POC tests, high specificity (and thus, low number of false positive test results) is most important. Dogs with a false positive result would not receive a re-vaccination, although possibly not being protected from infection at the time of the test, and this scenario has to be avoided. The CanTiCheck^®^ had the highest specificity and was very practicable. However, it has to be realized that differences regarding the number of false positive results of the CanTiCheck^®^ (one false positive), TiterCHEK^®^ (two false positives), and the ImmunoComb^®^ (three false positives) were minimal. The large variations in specificity between the POC tests in client-owned dogs from the present study are caused by the very low number of antibody-negative samples, which limits the possibility to assess the POC tests. However, this population mirrors exactly the epidemiological situation in which the tests are used in practice, as all dogs entering the hospital in which blood was drawn for various reasons were randomly included, and the aim of the study was to evaluate the POC tests under “normal” conditions in the field. The extremely high prevalence (97%) in client-owned dogs in the present study still was unexpected and is in the upper range of what has been reported in previous studies of neighboring areas (from 66% [14] up to 99% [5,20,21,22]). Such a high prevalence raises the question if it is required to test for anti-CPV antibodies at all, since nearly all dogs were protected. Possibly, the goal of a POC tests should therefore be to specifically find individual dogs that might still be unprotected, e.g., due to immunosuppression. However, as only very few antibody-negative dogs were present in the group of client-owned dogs, the sera from SPF dogs lacking antibodies against CPV were tested, to further evaluate specificity of the POC tests. Three of the POC tests (CanTiCheck^®^, TiterCHEK^®^, and ImmunoComb^®^) delivered no false positive results, thus confirming that all POC tests (except the Fastest^®^) had a high specificity.

So far, there are no data from independent studies on the CanTiCheck^®^. Diagnostic performance of the test in all dogs in the present study was good (specificity, 98%; sensitivity, 80%), but not as good as declared by the manufacturer (specificity: 99.9%; sensitivity: 98.1%) [23]. The TiterCHEK^®^ CDV/CPV has been evaluated by three independent studies in field dogs before [9,10,11]. The specificity of the TiterCHEK^®^ in the present study of the client-owned dog population (67%) was remarkably lower when compared to the former studies which examined samples of dogs from shelters and found specificities of 94% [11], 98% [9], and even 100% [10], with a prevalence of anti-CPV antibodies of 84% [11], 67% [9], and 85% [10], respectively. Specificity of the ImmunoComb^®^ was 81% in a study of Kim and colleagues (2017), compared to 50% in client-owned dogs in the present study. The FASTest^®^ has been evaluated in one study, and specificity was 94% with an anti-CPV antibody prevalence of 57% [12] compared to 33% in client-owned dogs in the present study. The reason for the lower specificities of the TiterCHEK^®^, the ImmunoComb^®^, and the FASTest^®^ in the present study, as compared to the previous studies, might be due to modifications to the tests by the manufacturer, with the aim of increasing the sensitivity of the tests [11]. Different batch qualities or even differences in the tests between the USA and Germany could also be possible explanations for these discrepancies. However, it also has to be considered that the specificities found in the present study do not necessarily differ markedly compared with those determined by previous studies, when taking the low number of VN-negative samples and thus the wide confidence intervals for specificity in the present study into account. When assessing specificities of the tests using the sera of all dogs (client-owned and SPF dogs), the results are more in line with the ones from the previous studies.

Other reasons for the discordance between the test results in the above mentioned studies [9,10,11], as well as the manufacturer´s declaration concerning diagnostic performance of the tests versus the results of the present study, could theoretically be invalid VN results (e.g., false negative results). However, the discrepancies between the POC tests themselves and absence of a good agreement in statistical analysis clearly imply false results in the POC tests and not in the VN (Table 7). In addition, there was not even one sample that was positive in all four POC tests and negative in VN. Moreover, the VN results were not solely based on cytopathic effects, as the VN assay was combined with an indirect immunofluorescence staining, which allows for the detection of CPV particles within the nuclei of infected cells, hence, avoiding invalid VN results.

In the present study, VN was used for measurement of anti-CPV antibodies. Hemagglutination inhibition (HI) can alternatively be performed. While VN is based on the cytopathic effect of parvoviruses, HI evaluates the antibodies´ ability to bind and agglutinate certain erythrocytes. In general, both tests are first-class correlated with protection, since they directly examine the biological effect of the antibodies [24] and are superior to other detection methods, such as the enzyme-linked immunosorbent assay or the indirect immunofluorescence test [25]. In comparison to HI, VN can be of advantage when testing samples containing CPV strains without hemagglutinating properties which have been described to occur [26]. In addition, VN is considered to have a higher sensitivity for detecting lower antibody titers and/or even antibodies after infection with new antigenic variants [27].

The PPV was high in all four POC tests in client-owned dogs, which was certainly influenced by the high antibody prevalence in the study population (Table 3). The PPV predicts the likelihood that a dog with a positive test result is antibody-positive and protected against disease and thus can be used to determine how reliably a positive result agrees with reality. However, it must be considered that the PPV is highly prevalence-dependent, and it decreases with decreasing antibody prevalence. In the present study, PPVs of the POC tests in client-owned dogs were nearly identical to the given antibody prevalence (97%). When testing with sera from SPF dogs only, three of the POC tests (CanTiCheck^®^, TiterCHEK^®^, and ImmunoComb^®^) had a specificity of 100%, and only the Fastest^®^ delivered false positive results.

On the other hand, the FASTest^®^ showed best results in sensitivity. A test lacking sensitivity means that protected dogs will have a negative test result and would be vaccinated unnecessarily, which is, however, not a severe problem. At the moment, this happens anyway for many dogs which receive the recommended triennial re-vaccinations (or even yearly re-vaccinations).

For use in veterinary practice, POC tests with the highest specificity (TiterCHEK^®^, CanTiCheck^®^, and ImmunoComb^®^) should be recommended. When comparing these tests with regard to other performance parameters, the CanTiCheck^®^ had a significantly higher sensitivity than the other tests and a better practicability; therefore, it should be considered to be the test of choice. Overall, however, modifications of all tests would be recommended to aim a higher specificity, even if this would lead to a lower sensitivity, which is not as important for the setting in which the test will be used.

The limitation of the study is the low number of antibody-negative samples in the client-owned dogs. To overcome this limitation, the sera of 43 SPF dogs were included, even though specific pathogen-free animals do not reflect the epidemiological conditions in the field. The use of VN as the reference standard in the present study, and not HI, makes the comparison between the results of the present study and those of previous studies that have used HI more difficult.

## 5. Conclusions

When considering specificity, which is most important for avoiding lack of vaccination in an unprotected dog, as well as practicability, the CanTiCheck^®^ would be the POC test of choice. Except for the FASTest^®^, differences regarding number of false positive results were minimal (using sera from client-owned dogs in the field) or absent (using sera from SPF dogs). Nevertheless, modifications of all tests would be recommended to aim for higher specificity.

## Figures and Tables

**Table 1 viruses-13-00018-t001:** Manufacturers’ instructions of the four point-of-care tests for detection of anti-canine parvovirus antibodies. Data on antibodies against canine distemper virus and canine adenovirus were not subjects of the present study.

Point-of-Care Test	ImmunoComb^®^ Canine VacciCheck, Enzyme-Linked Immunosorbent Assay	TiterCHEK^®^ CDV ^1^/CPV ^2^, Enzyme-Linked Immunosorbent Assay	FASTest^®^ CDV ^1^-CPV ^2^ Ab ^3^, Lateral Flow Immunoassay	CanTiCheck^®^, Lateral Flow Immunoassay
Antibodies	CPV ^2^, CDV ^1^, CAV ^4^	CPV ^2^, CDV ^1^	CPV ^2^, CDV ^1^	CPV ^2^, CDV ^1^
Storage	refrigerated	refrigerated	at room temperature	at room temperature
Application	after warming up to room temperature (60–120 min)	after warming up to room temperature (60–120 min)	immediately	immediately
Time of performance	21 min	20 min	10 min	10 min
Test material	serum (5 µL), plasma (5 µL), or whole blood (10 µL)	serum or plasma (each 1 µL)	serum, plasma, or whole blood (each 5 µL)	serum (50 µL)

^1^ CDV, canine distemper virus; ^2^ CPV, canine parvovirus; ^3^ Ab, antibodies; ^4^ CAV, canine adenovirus.

**Table 2 viruses-13-00018-t002:** Signalment and health status of the client-owned dogs.

Variable	Category	Number of Dogs
Breed	Mixed breed	47
	Purebred	151
Sex	Female	102
	Male	96
Neutering status	Intact	97
	Neutered	101
Health status	Healthy	22
	Immune-mediated disorders	1
	Endocrine disorders	8
	Inflammatory/infectious diseases	39
	Neoplasia	49
	Neurologic disease	18
	Cardiac disease	24
	Orthopedic problems	22
	Various other disorders	15

**Table 3 viruses-13-00018-t003:** Results of anti-canine parvovirus antibody testing in the sera of overall 241 dogs, in the subgroup of 198 client-owned dogs, and in the subgroup of 43 specific pathogen-free dogs, and comparison of the four point-of-care tests, using VN as the reference standard.

	ImmunoComb^®^ Negative	ImmunoComb^®^ Positive	TiterCHEK^®^ Negative	TiterCHEK^®^ Positive	FASTest^®^ Negative	FASTest^®^ Positive	CanTiCheck^®^ Negative	CanTiCheck^®^ Positive
**Results in sera of all 241 dogs (client-owned dogs and specific pathogen free dogs)**								
VN ^1^ negative (<10) (*n* = 49)	46 true negatives	3 false positives	47 true negatives	2 false positives	36 true negatives	13 false positives	48 true negatives	1 false positive
VN ^1^ positive (≥10) (*n* = 192)	57 false negatives	135 true positives	71 false negatives	121 true positives	9 false negatives	183 true positives	38 false negatives	154 true positives
**Results in sera of 198 client-owned dogs**								
VN ^1^ negative (<10) (*n* = 6)	3 true negatives	3 false positives	4 true negatives	2 false positives	2 true negatives	4 false positives	5 true negatives	1 false positives
VN ^1^ positive (≥10) (*n* = 192)	57 false negatives	135 true positives	71 false negatives	121 true positives	9 false negatives	183 true positives	38 false negatives	154 true positives
**Results in sera of 43 specific pathogen-free dogs**								
VN ^1^ negative (<10) (*n* = 43)	43 true negatives	0 false positives	43 true negatives	0 false positives	34 true negatives	9 false positives	43 true negatives	0 false positives
VN ^1^ positive (≥10) (*n* = 0)	0 false negatives	0 true positives	0 false negatives	0 true positives	0 false negatives	0 true positives	0 false negatives	0 true positives

^1^ VN, virus neutralization.

**Table 4 viruses-13-00018-t004:** Performance parameters of the four point-of-care tests to detect antibodies against canine parvovirus in all 241 dogs, in 198 client-owned dogs, and in 43 specific pathogen-free dogs, based on the results given in Table 1: sensitivity, specificity, positive predictive value, and negative predictive value, as well as overall accuracy, were calculated by using virus neutralization (VN) as the reference standard (considering a cutoff point of ≥10 as positive), at a given antibody prevalence of 80% in VN in the sera of all dogs, 97% in the sera of client-owned dogs, and 0% in the sera of specific pathogen-free dogs.

Point-of-Care Tests	ImmunoComb^®^	TiterCHEK^®^	FASTest^®^	CanTiCheck^®^
**Results in sera of all 241 dogs (client-owned dogs and specific pathogen free dogs)** **(prevalence of antibodies in virus neutralization: 80%)**				
Sensitivity ^1^ % (95% CI ^2^ %)	70 (63–77)	63 (56–70)	95 (91–98)	80 (74–86)
Specificity ^3^ % (95% CI ^2^ %)	94 (83–99)	96 (86–100)	73 (59–85)	98 (89–100)
Positive predictive value ^4^ % (95% CI ^2^ %)	98 (94–100)	98 (94–100)	93 (89–96)	99 (96–100)
Negative predictive value ^5^ % (95% CI ^2^ %)	45 (35–55)	40 (31–49)	80 (65–90)	56 (45–67)
Overall accuracy ^6^ % (95% CI ^2^ %)	75 (69–80)	70 (63–75)	91 (87–94)	84 (79–88)
**Results in sera of 198 client-owned dogs (prevalence of antibodies in virus neutralization: 97%)**				
Sensitivity ^1^ % (95% CI ^2^ %)	70 (63–77)	63 (56–70)	95 (91–98)	80 (74–86)
Specificity ^3^ % (95% CI ^2^ %)	50 (12–88)	67 (22–96)	33 (4–78)	83 (36–100)
Positive predictive value ^4^ % (95% CI ^2^ %)	98 (94–100)	98 (94–100)	98 (95–99)	99 (96–100)
Negative predictive value ^5^ % (95% CI ^2^ %)	5 (1–14)	5 (1–13)	18 (2–52)	12 (4–25)
Overall accuracy ^6^ % (95% CI ^2^ %)	70 (63–76)	63 (56–70)	93 (89–96)	80 (74–86)
**Results in sera of 43 specific pathogen free dogs (prevalence of antibodies in virus neutralization: 0%)**				
Sensitivity ^1^ % (95% CI ^2^ %)	n.d. ^7^	n.d. ^7^	n. d. ^7^	n.d. ^7^
Specificity ^3^ % (95% CI ^2^ %)	100	100	79 (64–90)	100
Positive predictive value ^4^ % (95% CI ^2^ %)	n.d. ^7^	n.d. ^7^	n.d. ^7^	n.d. ^7^
Negative predictive value ^5^ % (95% CI ^2^ %)	100	100	100	100
Overall accuracy ^6^ % (95% CI ^2^ %)	100	100	79 (64–90)	100

^1^ Sensitivity, true positive rate; ^2^ CI, confidence interval; ^3^ specificity, true negative rate; ^4^ positive predictive value, proportion of patients with positive test results in total of subjects with positive results; ^5^ negative predictive value, proportion of patients with negative test results in total of subjects with negative results; ^6^ overall accuracy, probability that a dog was correctly classified by the tests; ^7^ n.d., cannot be determined.

**Table 5 viruses-13-00018-t005:** Results of McNemar´s statistic to determine differences between specificity and sensitivity of the binary point-of-care tests in all dogs.

Point-of-Care Test	Specificity	McNemar´s *p*-Value for Comparison of Specificity	Sensitivity	McNemar´s *p*-Value for Comparison of Sensitivity
TiterCHEK^®^	96%	<0.001	63%	<0.001
FASTest^®^	73%		95%	
TiterCHEK^®^	96%	0.564	63%	0.099
ImmunoComb^®^	94%		70%	
FASTest^®^	73%	0.008	95%	<0.001
ImmunoComb^®^	94%		70%	
ImmunoComb^®^	94%	0.317	70%	0.008
CanTiCheck^®^	98%		80%	
CanTiCheck^®^	98%	0.317	80%	<0.001
TiterCHEK^®^	96%		63%	
CanTiCheck^®^	98%	<0.001	80%	<0.001
FASTest^®^	73%		95%	

**Table 6 viruses-13-00018-t006:** Agreement of the results of the four point-of-care tests, using Cohen´s kappa statistic in all dogs.

Point-of-Care Test	Agreement of Positive Results	Agreement of Negative Results	Kappa Coefficient	
			95% CI ^1^	*p*-Value
TiterCHEK^®^	121	43	0.35	<0.001
FASTest^®^			(0.23–0.47)	
TiterCHEK^®^	93	73	0.38	<0.001
ImmunoComb^®^			(0.26–0.49)	
FASTest^®^	132	39	0.36	<0.001
ImmunoComb^®^			(0.23–0.49)	
ImmunoComb^®^	119	67	0.52	<0.001
CanTiCheck^®^			(0.41–0.63)	
CanTiCheck^®^	111	74	0.53	<0.001
TiterCHEK^®^			(0.43–0.64)	
CanTiCheck^®^	153	43	0.54	<0.001
FASTest^®^			(0.43–0.66)	

^1^ CI, confidence interval.

**Table 7 viruses-13-00018-t007:** Subgroups of virus neutralization test results and the respective results of the four point-of-care tests for the detection of antibodies against canine parvovirus all dogs.

VN ^1^	ImmunoComb^®^	TiterCHEK^®^	FASTest^®^	CanTiCheck^®^	Number of Samples
+	+	+	+	+	87
+	+	+	+	-	4
+	-	+	+	-	6
+	-	-	+	-	12
+	-	-	-	-	4
+	+	-	+	-	9
+	-	-	+	+	12
+	-	+	-	+	1
+	+	-	-	+	1
+	+	+	-	-	1
+	+	-	+	+	31
+	+	-	-	-	2
+	-	+	+	+	22
-	-	-	+	-	11
-	+	+	+	-	1
-	+	-	-	-	2
-	-	+	+	+	1
-	-	-	-	-	34

^1^ VN, virus neutralization.

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
