# Peer review of "Comparison of Four Commercially Available Point-of-Care Tests to Detect Antibodies against Canine Parvovirus in Dogs"

_viruses, 2020, doi:10.3390/v13010018_

Round 1

Reviewer 1 Report

This remains an interesting manuscript that compared four point-of-care tests for detecting the presence of canine parvovirus antibodies in dogs. The major conclusion of the revised study is unchanged: CanTiCheck was the most specific and practical test when compared to FASTest, ImmunoComb, and TiterCHEK. The authors have clarified many points raised in the initial review that, in general, have strengthened the manuscript.

Author Response

Reviewer 1

Open Review

English language and style

( ) Extensive editing of English language and style required
( ) Moderate English changes required
(x) English language and style are fine/minor spell check required
( ) I don't feel qualified to judge about the English language and style

Yes

Can be improved

Must be improved

Not applicable

Does the introduction provide sufficient background and include all relevant references?

(x)

( )

( )

( )

Is the research design appropriate?

(x)

( )

( )

( )

Are the methods adequately described?

(x)

( )

( )

( )

Are the results clearly presented?

(x)

( )

( )

( )

Are the conclusions supported by the results?

(x)

( )

( )

( )

Comments and Suggestions for Authors

This remains an interesting manuscript that compared four point-of-care tests for detecting the presence of canine parvovirus antibodies in dogs. The major conclusion of the revised study is unchanged: CanTiCheck was the most specific and practical test when compared to FASTest, ImmunoComb, and TiterCHEK. The authors have clarified many points raised in the initial review that, in general, have strengthened the manuscript.

We thank the reviewer for the positive comments.

Spell checks were made throughout the manuscript.

Line 235: “…is most important.”

Line 256: “…which confirms that all POC tests (except the Fastest®) had a high specificity.

Line 230: “…in client-owned dogs,…”

Line 316: “…had a significantly higher sensitivity…”

Reviewer 2 Report

In a significant improvement to the previous manuscript, the authors have added more VN negative samples to the study, thus allowing assessment of the specificity of the tests in question. The new results are however as of yet incompletely integrated to the previous results, and the previous non-significant results are still expressed throughout the text in a manner that may easily mislead a busy reader. This warrants changes to the manuscript, but no additional experiments are needed. Thus I recommend accepting the manuscript with minor revisions. Please find below more specific comments.

To enhance the assessment of specificity of the tests, I strongly encourage the authors to combine the VN negative sample results of client-owned dogs and SPF dogs and calculate specificities from these samples grouped together, e.g. CanTiCheck: 43+5 =48 true negatives, 1 false positive -> specificity = 48/(48+1) = 98% (48/49). By analyzing these specificities instead of the ones acquired using only six VN negative samples, e.g. the discussion on lines 222-37, 245-61 and 298-303 could be improved (likely the composite specificities would be more in line of the previous studies as well), as well as other sections of the text. Also, I strongly encourage integrating the SPF dog data in the previous tables 3-7 instead of generating the new table 8.

Lines 24-30: Remove specificity figures, as they are mostly misleading (due to small number of VN negative samples from client-owned dogs). For clarity, add number of samples after each percentage in the abstract, e.g.: ”Prevalence of anti-CPV antibodies in client-owned dogs was 97% (192/198).”

Line 27: Remove "CanTiCheck® showed the highest specificity (not significantly different)." --> Combine the VN negative sample results from client-owned and SPF dogs to calculate specificity, as suggested above, and show these figures in the abstract instead of separate specificities from client-owned and SPF dogs, e.g. "Specificities of the tests were assessed by VN negative samples from 6 client-owned and 43 SPF dogs and were..."

Lines 31-2: Remove "However differences in the number of false positive results were minimal." Specificity of FASTest was worse than the other tests in the SPF dogs, and thus the differences in the number of false positives were not minimal.

Lines 67-9: Please add more information on how the SPF dog samples were selected, e.g. are these archival samples, originating from some other study, or what? Were these dogs previously vaccinated and the vaccine response had waned, or were they not at all vaccinated, or perhaps immunosuppressed? As already discussed by the authors later in the manuscript, a limitation of the SPF samples is that they may not be very representative of the VN negative dogs in the field, but to better assess this aspect more information would be helpful.

Line 119: ”0/43”, not ”0/0”.

Lines 131-4: "The CanTiCheck® and the TiterCHEK® were the tests with the highest specificities (83% and 67%, respectively); due to the overlapping CIs, 132 specificity of these tests was comparable and specificity of the CanTiCheck® did not differ significantly 133 from that of the TiterCHEK® (McNemars p-value: 0.317) (Table 6). " --> change to e.g. "No significant differences in specificity were observed between the tests due to small number of VN negative samples (Table 6).".

Line 147: add "." to the end.

Line 243: "except of the" --> "except the" and "very high" -> I would tone this down to "high". With just 43 samples tested, we can't really be sure whether the true specificity is e.g. 90% (high) or 99% (very high).

Line 300: significant -> significantly

Author Response

Reviewer 2

Open Review

English language and style

( ) Extensive editing of English language and style required
( ) Moderate English changes required
(x) English language and style are fine/minor spell check required
( ) I don't feel qualified to judge about the English language and style

Yes

Can be improved

Must be improved

Not applicable

Does the introduction provide sufficient background and include all relevant references?

(x)

( )

( )

( )

Is the research design appropriate?

(x)

( )

( )

( )

Are the methods adequately described?

( )

( )

(x)

( )

Are the results clearly presented?

( )

( )

(x)

( )

Are the conclusions supported by the results?

(x)

( )

( )

( )

We improved the methods according to the reviewer´s recommendations and presented the results accordingly (see below).

Comments and Suggestions for Authors

In a significant improvement to the previous manuscript, the authors have added more VN negative samples to the study, thus allowing assessment of the specificity of the tests in question. The new results are however as of yet incompletely integrated to the previous results, and the previous non-significant results are still expressed throughout the text in a manner that may easily mislead a busy reader. This warrants changes to the manuscript, but no additional experiments are needed. Thus I recommend accepting the manuscript with minor revisions. Please find below more specific comments.

We thank the reviewer for the comments. We integrated the results of the SPF dogs to those of the client-owned dogs and presented overall results according to the reviewer´s recommendations. Respective changes to the manuscript have been made according to the reviewer´s recommendations below. However, in our opinion the results from client-owned dogs should still be presented separately, since they correspond to the field situation in which the tests are used; furthermore, presentation of the results of client-owned dogs was endorsed by the other reviewers. Therefore, we decided to leave these results in the manuscript.

Abstract:

“Measuring antibodies to evaluate dogs´ immunity against canine parvovirus (CPV) is useful to avoid unnecessary re-vaccinations. The study aimed to evaluate the quality and practicability of four point-of-care (POC) tests for detection of anti-CPV antibodies. Sera of 198 client-owned and 43 specific pathogen-free (SPF) dogs were included; virus neutralization was used as reference method. Specificity, sensitivity, positive and negative predictive value (PPV, NPV), and overall accuracy (OA) were calculated. Specificity was considered as most important indicator for POC test performance. Differences between specificity and sensitivity of POC tests in sera of all dogs were determined by McNemar, agreement by Cohen´s kappa. Prevalence of anti-CPV antibodies in all dogs was 80% (192/241), in the subgroup of client-owned dogs 97% (192/198), in the subgroup of SPF dogs 0% (0/43). FASTest® and CanTiCheck® were easiest to perform. Specificities in all dogs and in only client-owned dogs were 94% and 50% (ImmunoComb®), 73% and 33% (FASTest®), 96% and 67% (TiterCHEK®), and 98% and 83% (CanTiCheck®), respectively. Specificity in SPF dogs was 100% for ImmunoComb®, TiterCHEK®, and CanTiCheck®, and 79% for FASTest®. FASTest® showed highest sensitivity (95%). No significant differences in specificity were observed between the ImmunoComb®, the TiterCHEK®, and the CanTiCheck®. CanTiCheck® would be the POC test of choice when considering specificity and practicability. However, differences in the number of false positive results between CanTiCheck®, TiterCHEK®, and ImmunoComb® were minimal.“

Line 123: “Results of the four POC tests were compared to the results of the VN assay for determination of the diagnostic quality of each assay. Sensitivity, specificity, PPV, NPV, and OA were calculated for all dogs, as well as for the subgroup of client-owned dogs, and for the subgroup of SPF dogs. Specificity was regarded as most important parameter for the quality of the tests. McNemar´s test was used for the paired study design to determine significant differences in specificity and sensitivity between the POC tests for all dogs. To assess agreement of the results among the four POC tests, Cohen´s Kappa statistic was used. P-values < 0.05 were considered significant.”

Line 133: “Prevalence of antibodies when considering sera of all dogs was 80% (192/241; 95% CI: 74-85). Antibodies against CPV were present in 97% (192/198; 95% CI: 93-99) of the client-owned dog population. None of the SPF dogs had anti-CPV antibodies (0/43; 95% CI: 0-10).”

Line 146: “Sensitivity, specificity, PPV, NPV, and OA of the four POC tests when compared to VN as reference standard in all groups of dogs are shown in Tables 3-4. Specificity was highest in the CanTiCheck® (overall dogs: 98%, client-owned dogs: 83%, SPF dogs: 100%) and the TiterCHEK® (overall dogs: 96%, client-owned dogs: 67%, SPF dogs: 100%); no significant differences in specificity were observed between the ImmunoComb®, the TiterCHEK®, and the CanTiCheck® when considering all dogs (Table 5).  Sensitivity was highest in the FASTest® (overall dogs: 95%, client-owned dogs: 95%) and the CanTiCheck® (overall dogs: 80%, client-owned dogs: 89%); sensitivity of the FASTest® was significantly higher compared to the other tests (McNemars p-value in each comparison: < 0.001), sensitivity of the CanTiCheck® was significantly higher compared to one of the TiterCHEK® (McNemars p-value: < 0.001).. Table 7 shows the agreement of the POC test results in all dogs. TiterCHEK® and FASTest® had the same result in 164/241 (68%) samples, TiterCHEK® and ImmunoComb® in 166/241 (69%), FASTest® and Immunocomb® in 180/241 (75%), CanTiCheck® and ImmunoComb® in 205/241 (85%), CanTiCheck® and TiterCHEK® in 219/241 (91%), CanTiCheck® and FASTest® in 213/241 (88%) samples.

Statistical analysis revealed a moderate agreement in the results of all dogs between the CanTiCheck® and TiterCHEK®: kappa = 0.53, between the CanTiCheck® and the ImmunoComb®: kappa = 0.52 and between CanTiCheck® and the FASTest®: kappa = 0.55. Agreement among the other POC tests was fair (Table 6). Table 7 shows the subgroups of virus neutralization test results and the respective results of the four point-of-care tests for the detection of antibodies aganist canine parvovirus in all dogs.

Table 3. Results of anti-canine parvovirus antibody testing in sera of overall 241 dogs, in the subgroup of of 198 client-owned dogs, and in the subgroup of 43 specific pathogen-free dogs, and comparison of the four point-of-care tests using VN as the reference standard.

ImmunoComb® negative

ImmunoComb® positive

TiterCHEK® negative

TiterCHEK® positive

FASTest® negative

FASTest® positive

CanTiCheck® negative

CanTiCheck® positive

Results in sera of all 241 dogs (client-owned dogs and specific pathogen free dogs)

VN 1 negative (< 10) (n=49)

46

true negatives

3

false positives

47

true negatives

2

false positives

36

true negatives

13

false positives

48

true negatives

1

false positives

VN 1 positive (≥ 10) (n=192)

57

false negatives

135

true positives

71

false negatives

121

true positives

9

false negatives

183

true positives

38

false negatives

154

true positives

Results in sera of 198 client-owned dogs

VN 1 negative (< 10) (n=6)

3

true negatives

3

false positives

4

true negatives

2

false positives

2

true negatives

4

false positives

5

true negatives

1

false positives

VN 1 positive (≥ 10) (n=192)

57

false negatives

135

true positives

71

false negatives

121

true positives

9

false negatives

183

true positives

38

false negatives

154

true positives

Results in sera of 43 specific pathogen-free dogs

VN 1 negative (< 10) (n=43)

43

true negatives

0

false positives

43

true negatives

0

false positives

34

true negatives

9

false positives

43

true negatives

0

false positives

VN 1 positive (≥ 10) (n=0)

0

false negatives

0

true positives

0

false negatives

0

true positives

0

false negatives

0

true positives

0

false negatives

0

true positives

1VN, virus neutralization

Table 4. Performance parameters of the four point-of-care tests to detect antibodies against canine parvovirus in all 241 dogs, in 198 client-owned dogs, and in 43 specific pathogen-free dogs based on results given in table 1: sensitivity, specificity, positive predictive value, and negative predictive value, as well as overall accuracy were calculated using virus neutralization (VN) as reference standard (considering a cut-off point of ≥10 as positive) at a given antibody prevalence of 80% in VN in sera of all dogs, 97% in sera of client-owned dogs, and 0% in sera of specific pathogen-free dogs.

Point-of-care tests

ImmunoComb®

TiterCHEK®

FASTest®

CanTiCheck®

Results in sera of all 241 dogs (client-owned dogs and specific pathogen free dogs) (prevalence of antibodies in virus neutralization: 80%)

Sensitivity  1 % (95% CI 2 %)

70 (63-77)

63 (56-70)

95 (91-98)

80 (74-86)

Specificity 3 % (95% CI 2 %)

94 (83-99)

96 (86-100)

73 (59-85)

98 (89-100)

Positive predictive value 4 % (95% CI 2 %)

98 (94-100)

98 (94-100)

93 (89-96)

99 (96-100)

Negative predictive value 5 % (95% CI 2 %)

45 (35-55)

40 (31-49)

80 (65-90)

56 (45-67)

Overall accuracy 6 % (95% CI 2 %)

75 (69-80)

70 (63-75)

91 (87-94)

84 (79-88)

Results in sera of 198 client-owned dogs (prevalence of antibodies in virus neutralization: 97%)

Sensitivity  1 % (95% CI 2 %)

70 (63-77)

63 (56-70)

95 (91-98)

80 (74-86)

Specificity 3 % (95% CI 2 %)

50 (12-88)

67 (22-96)

33 (4-78)

83 (36-100)

Positive predictive value 4 % (95% CI 2 %)

98 (94-100)

98 (94-100)

98 (95-99)

99 (96-100)

Negative predictive value 5 % (95% CI 2 %)

5 (1-14)

5 (1-13)

18 (2-52)

12 (4-25)

Overall accuracy 6 % (95% CI 2 %)

70 (63-76)

63 (56-70)

93 (89-96)

80 (74-86)

Results in sera of 43 specific pathogen free dogs (prevalence of antibodies in virus neutralization: 0%)

Sensitivity  1 % (95% CI 2 %)

n. d.7

n. d.7

n. d.7

n. d.7

Specificity 3 % (95% CI 2 %)

100

100

79 (64-90)

100

Positive predictive value 4 % (95% CI 2 %)

n. d.7

n. d.7

n. d.7

n. d.7

Negative predictive value 5 % (95% CI 2 %)

100

100

100

100

Overall accuracy 6 % (95% CI 2 %)

100

100

79 (64-90)

100

1 sensitivity, true positive rate; 2 CI, confidence interval; 3 specificity, true negative rate; 4 positive predictive value, proportion of patients with positive test results in total of subjects with positive results; 5 negative predictive value, proportion of patients with negative test results in total of subjects with negative results; 6 overall accuracy, probability that a dog was correctly classified by the tests; 7 n. d., cannot be determined

Table 5. Results of McNemar´s statistic to determine differences between specificity and sensitivity of the binary point-of-care tests in all dogs

Point-of-care test

Specificity

McNemar´s p-value

for comparison of specificity

Sensitivity

McNemar´s p-value

for comparison of sensitivity

TiterCHEK®

96%

<0.001

63%

< 0.001

FASTest®

73%

95%

TiterCHEK®

96%

0.564

63%

0.099

ImmunoComb®

94%

70%

FASTest®

73%

0.008

95%

< 0.001

ImmunoComb®

94%

70%

ImmunoComb®

94%

0.317

70%

0.008

CanTiCheck®

98%

80%

CanTiCheck®

98%

0.317

80%

< 0.001

TiterCHEK®

96%

63%

CanTiCheck®

98%

<0.001

80%

< 0.001

FASTest®

73%

95%

Table 7. Subgroups of virus neutralization test results and the respective results of the four point-of-care tests for the detection of antibodies aganist canine parvovirus all dogs

VN 1

ImmunoComb®

TiterCHEK®

FASTest®

CanTiCheck®

Number of samples

+

+

+

+

+

87

+

+

+

+

-

4

+

-

+

+

-

6

+

-

-

+

-

12

+

-

-

-

-

4

+

+

-

+

-

9

+

-

-

+

+

12

+

-

+

-

+

1

+

+

-

-

+

1

+

+

+

-

-

1

+

+

-

+

+

31

+

+

-

-

-

2

+

-

+

+

+

22

-

-

-

+

-

11

-

+

+

+

-

1

-

+

-

-

-

2

-

-

+

+

+

1

-

-

-

-

-

34

1 VN, virus neutralization

Line 237: “The CanTiCheck® had the highest specificity and was very practicable. However, it has to be realized that differences regarding the number of false positive results of the CanTiCheck® (one false positive), TiterCHEK® (two false positives), and the ImmunoComb® (three false positives) were minimal. The large variations in specificity between the POC tests in client-owned dogs from the present study are caused by the very low number of CPV antibody-negative samples, which limits the possibility to assess the POC tests.”

Line 258: “So far, there are no data from independent studies on the CanTiCheck®. Diagnostic performance of the test in all dogs in the present study was good (specificity: 98%; sensitivity: 80%)…”

Line 276: “When assessing specificities of the tests using serum of all dogs (client-owned and SPF dogs), results are more in line with the ones from the previous studies.”

Line 314: “For use in veterinary practice, the POC tests with the highest specificity (TiterCHEK®, CanTiCheck®, and Immunocomb®) should be recommended.”

To enhance the assessment of specificity of the tests, I strongly encourage the authors to combine the VN negative sample results of client-owned dogs and SPF dogs and calculate specificities from these samples grouped together, e.g. CanTiCheck: 43+5 =48 true negatives, 1 false positive -> specificity = 48/(48+1) = 98% (48/49).

We combined the samples and calculated the performance parameters for overall all dogs (Table 3 and 4). However, in our opinion the results from client-owned dogs should still be presented separately, since they correspond to the field situation in which the tests are used; furthermore, presentation of the results of client-owned dogs was endorsed by the other reviewers. Therefore, we decided to leave these results in the manuscript (respective changes see above).

By analyzing these specificities instead of the ones acquired using only six VN negative samples, e.g. the discussion on lines 222-37, 245-61 and 298-303 could be improved (likely the composite specificities would be more in line of the previous studies as well), as well as other sections of the text.

The discussion was changed accordingly.

Line 258: “So far, there are no data from independent studies on the CanTiCheck®. Diagnostic performance of the test in all dogs in the present study was good (specificity: 98%; sensitivity: 80%)…”

Line 276: “When assessing specificities of the tests using serum of all dogs (client-owned and SPF dogs), results are more in line with the ones from the previous studies.”

Line 314: “For use in veterinary practice, the POC tests with the highest specificity (TiterCHEK®, CanTiCheck®, and Immunocomb®) should be recommended.”

Also, I strongly encourage integrating the SPF dog data in the previous tables 3-7 instead of generating the new table 8.

Results of the SPF dogs were integrated in the previous tables (table 3-6) and table 8 was deleted (table see above). Statistical analyses were performed with results from all dogs only.

Line 146: “Sensitivity, specificity, PPV, NPV, and OA of the four POC tests when compared to VN as reference standard in all groups of dogs are shown in Tables 3-4. Specificity was highest in the CanTiCheck® (overall dogs: 98%, client-owned dogs: 83%, SPF dogs: 100%) and the TiterCHEK® (overall dogs: 96%, client-owned dogs: 67%, SPF dogs: 100%); no significant differences in specificity were observed between the ImmunoComb®, the TiterCHEK®, and the CanTiCheck® when considering all dogs (Table 5).  Sensitivity was highest in the FASTest® (overall dogs: 95%, client-owned dogs: 95%) and the CanTiCheck® (overall dogs: 80%, client-owned dogs: 89%); sensitivity of the FASTest® was significantly higher compared to the other tests (McNemars p-value in each comparison: < 0.001), sensitivity of the CanTiCheck® was significantly higher compared to one of the TiterCHEK® (McNemars p-value: < 0.001).. Table 7 shows the agreement of the POC test results in all dogs. TiterCHEK® and FASTest® had the same result in 164/241 (68%) samples, TiterCHEK® and ImmunoComb® in 166/241 (69%), FASTest® and Immunocomb® in 180/241 (75%), CanTiCheck® and ImmunoComb® in 205/241 (85%), CanTiCheck® and TiterCHEK® in 219/241 (91%), CanTiCheck® and FASTest® in 213/241 (88%) samples.

Statistical analysis revealed a moderate agreement in the results of all dogs between the CanTiCheck® and TiterCHEK®: kappa = 0.53, between the CanTiCheck® and the ImmunoComb®: kappa = 0.52 and between CanTiCheck® and the FASTest®: kappa = 0.55. Agreement among the other POC tests was fair (Table 6). Table 7 shows the subgroups of virus neutralization test results and the respective results of the four point-of-care tests for the detection of antibodies aganist canine parvovirus in all dogs.

Further changes that have been made: Table 5 has changed to table 7.

Lines 24-30: Remove specificity figures, as they are mostly misleading (due to small number of VN negative samples from client-owned dogs).

In our opinion, the results from client-owned dogs should still be presented separately, since they correspond to the field situation in which the tests are used; furthermore, presentation of the results of client-owned dogs was endorsed by the other reviewers. Therefore, we decided to leave these results in the manuscript.

Line 24: “Prevalence of anti-CPV antibodies in all dogs was 80% (192/241), in the subgroup of client-owned dogs 97% (192/198), in the subgroup of SPF dogs 0% (0/43). FASTest® and CanTiCheck® were easiest to perform. Specificities in all dogs and in only client-owned dogs were 94% and 50% (ImmunoComb®), 73% and 33% (FASTest®), 96% and 67% (TiterCHEK®), and 98% and 83% (CanTiCheck®), respectively. Specificity in SPF dogs was 100% for ImmunoComb®, TiterCHEK®, and CanTiCheck®, and 79% for FASTest®. FASTest® showed highest sensitivity (95%). No significant differences in specificity were observed between the ImmunoComb®, the TiterCHEK®, and the CanTiCheck®. CanTiCheck® would be the POC test of choice when considering specificity and practicability. However, differences in the number of false positive results between CanTiCheck®, TiterCHEK®, and ImmunoComb® were minimal.“For clarity, add number of samples after each percentage in the abstract, e.g.: ”Prevalence of anti-CPV antibodies in client-owned dogs was 97% (192/198).”

We added the number of samples.

Line 24: “Prevalence of anti-CPV antibodies in all dogs was 80% (192/241), in the subgroup of client-owned dogs 97% (192/198), in the subgroup of SPF dogs 0% (0/43).

Line 27: Remove "CanTiCheck® showed the highest specificity (not significantly different)."

We removed the sentence.

--> Combine the VN negative sample results from client-owned and SPF dogs to calculate specificity, as suggested above, and show these figures in the abstract instead of separate specificities from client-owned and SPF dogs, e.g. "Specificities of the tests were assessed by VN negative samples from 6 client-owned and 43 SPF dogs and were..."

We presented the results of the combined samples in addition to the results of the client-owned dogs (see above).

Lines 31-2: Remove "However differences in the number of false positive results were minimal." Specificity of FASTest was worse than the other tests in the SPF dogs, and thus the differences in the number of false positives were not minimal.

We changed the sentence.

Line 32: “However, differences in the number of false positive results between CanTiCheck®, TiterCHEK® , and ImmunoComb® were minimal. “

Lines 67-9: Please add more information on how the SPF dog samples were selected, e.g. are these archival samples, originating from some other study, or what? Were these dogs previously vaccinated and the vaccine response had waned, or were they not at all vaccinated, or perhaps immunosuppressed? As already discussed by the authors later in the manuscript, a limitation of the SPF samples is that they may not be very representative of the VN negative dogs in the field, but to better assess this aspect more information would be helpful.

The serum samples from the SPF dogs originated from a previous study, and were stored at -80°C. They samples derived from SPF dogs that have never been vaccinated. We added the information to the material and methods.

Line 78: “In addition, 43 samples of specific pathogen-free (SPF) dogs that had never been vaccinated were included in the study. These stored samples were provided by the Institute of Animal Hygiene and Veterinary Public Health, University of Leipzig, and the Institute for Infectious Diseases and Zoonoses, LMU Munich. All samples were stored at -80°C until testing.“

Line 119: ”0/43”, not ”0/0”.

We thank the reviewer for making us aware of this mistake.

Line 135: “None of the SPF dogs had anti-CPV antibodies (0/43; 95% CI: 0-10).”

Lines 131-4: "The CanTiCheck® and the TiterCHEK® were the tests with the highest specificities (83% and 67%, respectively); due to the overlapping CIs, 132 specificity of these tests was comparable and specificity of the CanTiCheck® did not differ significantly 133 from that of the TiterCHEK® (McNemars p-value: 0.317) (Table 6). " --> change to e.g. "No significant differences in specificity were observed between the tests due to small number of VN negative samples (Table 6).".

We made the analysis for all dogs. The sentence was changed according to the new results

Line 148: “Specificity was highest in the CanTiCheck® (overall dogs: 98%, client-owned dogs: 83%, SPF dogs: 100%) and the TiterCHEK® (overall dogs: 96%, client-owned dogs: 67%, SPF dogs: 100%); no significant differences in specificity were observed between the ImmunoComb®, the TiterCHEK®, and the CanTiCheck® when considering all dogs (Table 5).

Line 147: add "." to the end.

We deleted the sentence.

Line 243: "except of the" --> "except the" and "very high" -> I would tone this down to "high". With just 43 samples tested, we can't really be sure whether the true specificity is e.g. 90% (high) or 99% (very high).

We made the changes according to the reviewer´s recommendations.

Line 256: “…which confirms that all POC tests (except the Fastest®) had a high specificity.

Line 300: significant -> significantly

We made the correction.

Line 300: “…had a significantly higher sensitivity…”

Further spell checks were made throughout the manuscript.

Line 235: “…is most important.”

Line 256: “…which confirms that all POC tests (except the Fastest®) had a high specificity.

Line 230: “…in client-owned dogs,…”

Reviewer 3 Report

General comments

The paper viruses-1049707 entitled “Comparison of four commercially available point-of-care tests to detect antibodies against canine parvovirus in dogs” is a very well written and readable paper.

The material and methods are clear and well explained.

Minor mistakes:

In statistical analysis section, an explanation of the reason why sera of SPF animals were not included in the same analysis of the client-owned dogs should be added. The reason is reported at lines 306-307 but should be explained in material and methods.

Moreover, sensitivity, specificity, PPV, NPV and OA were calculated also for SPF animals.

Table 2. It would be helpful that the percentage of the population should be added alongside the absolute number in the column “Number of dogs”.

Line 230: “in client-owned dogs” is written in smaller characters.

Author Response

Reviewer 3

Open Review

English language and style

( ) Extensive editing of English language and style required
( ) Moderate English changes required
(x) English language and style are fine/minor spell check required
( ) I don't feel qualified to judge about the English language and style

Yes

Can be improved

Must be improved

Not applicable

Does the introduction provide sufficient background and include all relevant references?

(x)

( )

( )

( )

Is the research design appropriate?

(x)

( )

( )

( )

Are the methods adequately described?

(x)

( )

( )

( )

Are the results clearly presented?

(x)

( )

( )

( )

Are the conclusions supported by the results?

(x)

( )

( )

( )

Comments and Suggestions for Authors

General comments

The paper viruses-1049707 entitled “Comparison of four commercially available point-of-care tests to detect antibodies against canine parvovirus in dogs” is a very well written and readable paper. The material and methods are clear and well explained.

We thank the reviewer for the positive comments.

Minor mistakes:

In statistical analysis section, an explanation of the reason why sera of SPF animals were not included in the same analysis of the client-owned dogs should be added. The reason is reported at lines 306-307 but should be explained in material and methods.

According to the recommendation of reviewer 2, sera of the SPF dogs were included in the analysis.

Moreover, sensitivity, specificity, PPV, NPV and OA were calculated also for SPF animals.

Table 2. It would be helpful that the percentage of the population should be added alongside the absolute number in the column “Number of dogs”.

Line 230: “in client-owned dogs” is written in smaller characters.

We made the correction.

Line 230: “…in client-owned dogs,…”

Further spell checks were made throughout the manuscript.

Line 235: “…is most important.”

Line 256: “…which confirms that all POC tests (except the Fastest®) had a high specificity.

Line 316: “…had a significantly higher sensitivity…”

This manuscript is a resubmission of an earlier submission. The following is a list of the peer review reports and author responses from that submission.

Round 1

Reviewer 1 Report

Bergmann et al compared four point-of-care (POC) tests for detecting the presence of canine parvovirus antibodies in dogs. They evaluated the specificity, sensitivity, positive and negative predictive values, plus the overall accuracy of POC tests from 198 dog sera that were used for neutralization assays. The authors concluded that CanTiCheck was the most specific and practical POC test when compared to FASTest, ImmunoComb, and TiterCHEK. The manuscript is well written and the suggestions here will contribute to better virus antibody monitoring and help reduce unnecessary re-vaccinations of dogs.

Author Response

Reviewer 1

Open Review

English language and style

( ) Extensive editing of English language and style required
( ) Moderate English changes required
(x) English language and style are fine/minor spell check required
( ) I don't feel qualified to judge about the English language and style

Language corrections have been made throughout the manuscript.

Line 17: “Specificity, sensitivity, positive predictive value (PPV), negative predictive value (NPV), and overall accuracy (OA) were determined. Differences between specificity and sensitivity of the POC tests were determined by McNemar´s test, agreement among the POC tests by Cohen´s kappa.”

Virus neutralization (VN) detects antibodies that neutralize infectious particles and prevent infection [8]. VN can be used to determine the magnitude of CPV antibody titres but it can only be performed in specialized laboratories [8].”

Line 51: “…has been evaluated by…”

Line 53: “…has been evaluated by…”

Line 56: “The aim of this study was to compare the four commercially available POC tests detecting antibodies against CPV concerning their practicability, specificity, sensitivity, positive predictive value (PPV), negative predictive value (NPV), as well as overall accuracy (OA) using VN as the reference standard.“

Line 87: “…overnight…”

Line 89: “...from a privately-owned vaccinated…“

Line 98: “The ImmunoComb® and the TiterCHEK® are enzyme-linked immunosorbent assays”

Line 123: “Sensitivity, specificity, PPV, NPV, and OA of…”

Line 125: “…were the tests with the highest specificities (83% and 67%, respectively)…”

Line 129:”… were the tests with the highest sensitivities (95% and 80%, respectively)…”

Line 135: “Statistical analysis revealed poor agreement between TiterCHEK® and FASTest®: kappa=0.124, FASTest® and ImmunoComb®: kappa=0.052, TiterCHEK® and ImmunoComb®: kappa=0.162, CanTiCheck® and ImmunoComb®: kappa=0.285, CanTiCheck® and FASTest®: kappa=0.052 (Table 6). There was a minimal agreement between CanTiCheck® and TiterCHEK®: kappa=0.344.”

Line 140: Table 2. Results of canine parvovirus antibody testing in sera of 198 dogs and comparison of the four point-of-care tests using VN as the reference standard

Line 152: “Table 3. Performance parameters of the four point-of-care tests to detect antibodies against canine parvovirus based on results given in table 1: sensitivity, specificity, positive predictive value, and negative predictive value,…”

Line 161: “overall accuracy, probability that a dog was correctly classified by the tests”

Line 184: “Table 6. Agreement of the results of the four point-of-care tests using Cohen´s kappa statistic”

Line 190: “Although disease caused by CPV infection is rare in Northern European countries today, the risk of spread of CPV via import of dogs from Southern or Eastern Europe is still high [17,18].”

Line 203: “both tests were unproblematic to perform.”

Line 204: “…buffered-mixed…”

Line 214: “…realized…”

Line 237: “The FASTest® has been evaluated…”

Line 258:” Hemagglutination inhibition (HI)“

Line 264: ”…without hemagglutinating properties…”

Yes

Can be improved

Must be improved

Not applicable

Does the introduction provide sufficient background and include all relevant references?

(x)

( )

( )

( )

Is the research design appropriate?

(x)

( )

( )

( )

Are the methods adequately described?

(x)

( )

( )

( )

Are the results clearly presented?

(x)

( )

( )

( )

Are the conclusions supported by the results?

(x)

( )

( )

( )

Comments and Suggestions for Authors

Bergmann et al compared four point-of-care (POC) tests for detecting the presence of canine parvovirus antibodies in dogs. They evaluated the specificity, sensitivity, positive and negative predictive values, plus the overall accuracy of POC tests from 198 dog sera that were used for neutralization assays. The authors concluded that CanTiCheck was the most specific and practical POC test when compared to FASTest, ImmunoComb, and TiterCHEK. The manuscript is well written and the suggestions here will contribute to better virus antibody monitoring and help reduce unnecessary re-vaccinations of dogs.

We thank the reviewer for the positive comments.

Reviewer 2 Report

In the manuscript ”Comparison of four commercially available point-of-care tests to detect antibodies against canine parvovirus in dogs”, the authors comparatively evaluate the performance of four commercially available point-of-care canine parvovirus antibody assays against virus neutralization. The evaluation was performed using a panel of 198 dog sera collected at an university clinic, including 6 virus neutralization (VN) negative and 192 VN positive samples.

To reliably evaluate commercial diagnostic assays, independent comparative studies such as the one authors present are highly important. The authors had designed the sample collection to achieve enough virus neutralization negative samples for reliable assessment of assay specificity. Regrettably, there were far fewer VN negative samples than expected.

As pointed out by the authors, specificity is of utmost importance in evaluation of these diagnostic assays to avoid a false sense of protection from infection due to false positive antibody test results. To reliably assess the specificity differences between tests, more VN negative samples need to be included, as the authors initially had planned to.

I thus consider the data insufficient to warrant publication before more VN negative samples are studied, even though the study otherwise was well designed and the manuscript clear and concise.

More specific comments:

Table 4: For better readability, I would suggest removing rows with zero samples. Furthermore, I would remove colors from this table. To make comparison between tables easier, I suggest placing the tests in the same column order in tables 3 and 4.

In tables 3, 5 and 6 expressions ”Tests”, ”point-of-care tests” and ”point-of-care test” are used depending on the table. I would unify this to e.g. ”point-of-care test” for all of these tables.

Lines 195-197: ”Detection of CPV antibodies (at any level) is predictive for protection in adult dogs that have been vaccinated or infected previously [19]”

However, reference 19 is a paper on CDV (canine distemper virus), not CPV (canine parvovirus)? Please clarify / correct.

Line 209: ”both tests were unproblematically to perform” → unproblematic?

Lines 214-215: ”For assessing performance of the POC tests, high specificity (and thus, low number of false positive test results) are most important.”

For POC tests used in e.g. screening of disease, high sensitivity can be the first priority. Although it is implicated by the context, please specify in this sentence that you mean CPV antibody POC tests and not POC tests in general.

Lines 217-218: ”However, it has to be realized that differences regarding the number of false positive results of the POC tests were minimal (CanTiCheck ® : one false positive; TiterCHEK ® : two false positives; ImmunoComb ® : three false positives; FASTest ® : four false positives) and statistically not significant.”

There were only 6 virus neutralization negative samples in the sample panel, making it impossible to generate data with statistically significant differences in specificity between the tests. For this reason I consider it not worth mentioning and possibly even somewhat misleading to note that the differences were statistically not significant.

Lines 235-238: ”The TiterCHEK ® CDV/CPV has been evaluated by three independent studies before [9 – 11]. The specificity of the TiterCHEK ® in the present study (67%) was remarkably lower in comparison to the former studies which examined samples of dogs from shelters and found specificities of 94% [11], 98% [9], and even 100% [10] with a prevalence of CPV antibodies of 84% [11], 67% [9], and 85% [10], respectively”

Considering the low number of VN negative samples in the present study and thus the wide confidence interval for specificity (22-96%), the specificity found in the present study is not necessarily in conflict with the previous studies.

Furthermore, it could be considered whether performing haemagglutination inhibition assays alongside VN is feasible. This would allow more reliable comparison between this and previous studies as well as strenghten the evidence base on upon which to evaluate the POC assay performance. I do not consider this strictly necessary for publication, however.

Author Response

Reviewer 2

Open Review

English language and style

( ) Extensive editing of English language and style required
( ) Moderate English changes required
(x) English language and style are fine/minor spell check required
( ) I don't feel qualified to judge about the English language and style

Language corrections have been made throughout the manuscript.

Line 17: “Specificity, sensitivity, positive predictive value (PPV), negative predictive value (NPV), and overall accuracy (OA) were determined. Differences between specificity and sensitivity of the POC tests were determined by McNemar´s test, agreement among the POC tests by Cohen´s kappa.”

Virus neutralization (VN) detects antibodies that neutralize infectious particles and prevent infection [8]. VN can be used to determine the magnitude of CPV antibody titres but it can only be performed in specialized laboratories [8].”

Line 51: “…has been evaluated by…”

Line 53: “…has been evaluated by…”

Line 56: “The aim of this study was to compare the four commercially available POC tests detecting antibodies against CPV concerning their practicability, specificity, sensitivity, positive predictive value (PPV), negative predictive value (NPV), as well as overall accuracy (OA) using VN as the reference standard.“

Line 87: “…overnight…”

Line 89: “...from a privately-owned vaccinated…“

Line 98: “The ImmunoComb® and the TiterCHEK® are enzyme-linked immunosorbent assays”

Line 123: “Sensitivity, specificity, PPV, NPV, and OA of…”

Line 125: “…were the tests with the highest specificities (83% and 67%, respectively)…”

Line 129:”… were the tests with the highest sensitivities (95% and 80%, respectively)…”

Line 135: “Statistical analysis revealed poor agreement between TiterCHEK® and FASTest®: kappa=0.124, FASTest® and ImmunoComb®: kappa=0.052, TiterCHEK® and ImmunoComb®: kappa=0.162, CanTiCheck® and ImmunoComb®: kappa=0.285, CanTiCheck® and FASTest®: kappa=0.052 (Table 6). There was a minimal agreement between CanTiCheck® and TiterCHEK®: kappa=0.344.”

Line 140: Table 2. Results of canine parvovirus antibody testing in sera of 198 dogs and comparison of the four point-of-care tests using VN as the reference standard

Line 152: “Table 3. Performance parameters of the four point-of-care tests to detect antibodies against canine parvovirus based on results given in table 1: sensitivity, specificity, positive predictive value, and negative predictive value,…”

Line 161: “overall accuracy, probability that a dog was correctly classified by the tests”

Line 184: “Table 6. Agreement of the results of the four point-of-care tests using Cohen´s kappa statistic”

Line 190: “Although disease caused by CPV infection is rare in Northern European countries today, the risk of spread of CPV via import of dogs from Southern or Eastern Europe is still high [17,18].”

Line 203: “both tests were unproblematic to perform.”

Line 204: “…buffered-mixed…”

Line 214: “…realized…”

Line 237: “The FASTest® has been evaluated…”

Line 258:” Hemagglutination inhibition (HI)“

Line 264: ”…without hemagglutinating properties…”

Yes

Can be improved

Must be improved

Not applicable

Does the introduction provide sufficient background and include all relevant references?

( )

(x)

( )

( )

Is the research design appropriate?

( )

( )

(x)

( )

Are the methods adequately described?

(x)

( )

( )

( )

Are the results clearly presented?

( )

(x)

( )

( )

Are the conclusions supported by the results?

( )

( )

(x)

( )

Comments and Suggestions for Authors

In the manuscript ”Comparison of four commercially available point-of-care tests to detect antibodies against canine parvovirus in dogs”, the authors comparatively evaluate the performance of four commercially available point-of-care canine parvovirus antibody assays against virus neutralization. The evaluation was performed using a panel of 198 dog sera collected at an university clinic, including 6 virus neutralization (VN) negative and 192 VN positive samples.

To reliably evaluate commercial diagnostic assays, independent comparative studies such as the one authors present are highly important.

We thank the reviewer for this positive comment.

The authors had designed the sample collection to achieve enough virus neutralization negative samples for reliable assessment of assay specificity. Regrettably, there were far fewer VN negative samples than expected.

It is correct that the authors expected to have more VN negative samples. However, the aim of the study was to evaluate the POC tests under “normal” conditions in the field. Thus, the authors intentionally planned a sample collection which is in line with the realistic situation and the situation in which the tests will be used; dogs were randomly included. We already discussed that fact in the manuscript; we further added this point to the limitations and to the conclusions.

Line 220-226: “The large variations in specificity between the POC tests in the present study are caused by the very low number of CPV antibody-negative samples, which of course limits the possibility to assess the POC tests. However, the present study mirrors exactly the epidemiological situation in which the tests are used in practice as all dogs entering the hospital in which blood was drawn for various reasons were randomly included, and the aim of the study was to evaluate the POC tests under “normal” conditions in the field.”

As pointed out by the authors, specificity is of utmost importance in evaluation of these diagnostic assays to avoid a false sense of protection from infection due to false positive antibody test results. To reliably assess the specificity differences between tests, more VN negative samples need to be included, as the authors initially had planned to. I thus consider the data insufficient to warrant publication before more VN negative samples are studied, even though the study otherwise was well designed and the manuscript clear and concise.

To find antibody-negative dogs, specific pathogen-free animals would have to be included but even those are likely vaccinated and thus have CPV antibodies. The study aimed to evaluate the tests under field conditions. A case-control sampling is not feasible for such a study design. In addition, we do not have additional tests from the same batches available anymore. Given that, the authors hope that the manuscript can be considered for publication. The results are especially important because all test manufacturers should aim to improve specificity since any false positive result might be fatal. We added the reviewer´s concerns to the discussion.

The conclusion has also been changed according to the reviewer´s concerns.

Line 297:

“However, the low number of CPV antibody-negative samples limited the assessment of the POC tests, Nevertheless, modifications of all tests would be recommended to aim for higher specificity.”

More specific comments:

Table 4: For better readability, I would suggest removing rows with zero samples. Furthermore, I would remove colors from this table. To make comparison between tables easier, I suggest placing the tests in the same column order in tables 3 and 4.

We made the changes according to the reviewer´s comments. We removed the rows with zero samples and the colors from table 4.

Line 176:

“Table 4. Cross-classified test results of virus neutralization and the four point-of-care tests for the detection of antibodies aganist canine parvovirus

VN 1

ImmunoComb®

TiterCHEK®

FASTest®

CanTiCheck®

number of samples

+

+

+

+

+

87

+

+

+

+

-

4

+

-

+

+

-

6

+

-

-

+

-

12

+

-

-

-

-

4

+

+

-

+

-

9

+

-

-

+

+

12

+

-

+

-

+

1

+

+

-

-

+

1

+

+

+

-

-

1

+

+

-

+

+

31

+

+

-

-

-

2

+

-

+

+

+

22

-

-

-

+

-

2

-

+

+

+

-

1

-

+

-

-

-

2

-

-

+

+

+

1

1 VN, virus neutralization“

We replaced the tests in the same order in table 2-4.

Line 140: “Table 2” - Order: ImmunoComb® TiterCHEK® FASTest® CanTiCheck®

Line 152: “Table 3” - Order: ImmunoComb® TiterCHEK® FASTest® CanTiCheck®

Line 173: “Table 4” - Order: ImmunoComb® TiterCHEK® FASTest® CanTiCheck®

In tables 3, 5 and 6 expressions ”Tests”, ”point-of-care tests” and ”point-of-care test” are used depending on the table. I would unify this to e.g. ”point-of-care test” for all of these tables.

We thank the reviewer for making us aware of this mistake. The expression “Point-of-care test” was used for the respective tables.

Line 152 (Table 3): “Point-of-care test”

Line 181 (Table 5): “Point-of-care test”

Line 184 (Table 6): “Point-of-care test”

Lines 195-197: ”Detection of CPV antibodies (at any level) is predictive for protection in adult dogs that have been vaccinated or infected previously [19]”

However, reference 19 is a paper on CDV (canine distemper virus), not CPV (canine parvovirus)? Please clarify / correct.

We thank the reviewer for making us aware of this mistake. We added the correct reference.

Line 335: „Schultz, R.D.; Thiel, B.; et al. Age and long-term protective immunity in dogs and cats. J Comp Path 2010, 142, 102-108.”

Line 209: ”both tests were unproblematically to perform” → unproblematic?

We made the correction.

Line 203: “both tests were unproblematic to perform.”

Lines 214-215: ”For assessing performance of the POC tests, high specificity (and thus, low number of false positive test results) are most important.”

For POC tests used in e.g. screening of disease, high sensitivity can be the first priority. Although it is implicated by the context, please specify in this sentence that you mean CPV antibody POC tests and not POC tests in general.

We specified “POC tests” in the sentence.

Line 210: “For assessing performance of the pre-vaccination antibody POC tests, high specificity…”

Lines 217-218: ”However, it has to be realized that differences regarding the number of false positive results of the POC tests were minimal (CanTiCheck ® : one false positive; TiterCHEK ® : two false positives; ImmunoComb ® : three false positives; FASTest ® : four false positives) and statistically not significant.”

There were only 6 virus neutralization negative samples in the sample panel, making it impossible to generate data with statistically significant differences in specificity between the tests. For this reason I consider it not worth mentioning and possibly even somewhat misleading to note that the differences were statistically not significant.

Statistical evaluation of data still gives valuable information. If for example a POC test would have had 6 false positive results and another no false positive results, this would have been a significant finding. However, we appreciate the comment of the reviewer and deleted the statement.

Line 214: “However, it has to be realized that differences regarding the number of false positive results of the POC tests were minimal (CanTiCheck®: one false positive; TiterCHEK®: two false positives; ImmunoComb®: three false positives; FASTest®: four false positives).”

Lines 235-238: ”The TiterCHEK ® CDV/CPV has been evaluated by three independent studies before [9 – 11]. The specificity of the TiterCHEK ® in the present study (67%) was remarkably lower in comparison to the former studies which examined samples of dogs from shelters and found specificities of 94% [11], 98% [9], and even 100% [10] with a prevalence of CPV antibodies of 84% [11], 67% [9], and 85% [10], respectively”

Considering the low number of VN negative samples in the present study and thus the wide confidence interval for specificity (22-96%), the specificity found in the present study is not necessarily in conflict with the previous studies.

We agree with the reviewer´s comment and added this point to the discussion.

Furthermore, it could be considered whether performing haemagglutination inhibition assays alongside VN is feasible. This would allow more reliable comparison between this and previous studies as well as strenghten the evidence base on upon which to evaluate the POC assay performance. I do not consider this strictly necessary for publication, however.

The authors deliberately decided to use the VN and not the HI as reference standard according to the reasons listed in the discussion. However, we appreciate the comment of the reviewer and discussed that point accordingly.

Reviewer 3 Report

The paper entitled "Comparison of four commercially available point-of-care tests to detect antibodies against canine parvovirus in dogs" is a very useful, well written, and easily readable study.

It will have a significant impact in practice.

I only found two typos:

line 208: buffered instead of bufferd

line 218: realized instead of realzied.

Author Response

Reviewer 3

Open Review

English language and style

( ) Extensive editing of English language and style required
( ) Moderate English changes required
(x) English language and style are fine/minor spell check required
( ) I don't feel qualified to judge about the English language and style

Language corrections have been made throughout the manuscript.

Line 17: “Specificity, sensitivity, positive predictive value (PPV), negative predictive value (NPV), and overall accuracy (OA) were determined. Differences between specificity and sensitivity of the POC tests were determined by McNemar´s test, agreement among the POC tests by Cohen´s kappa.”

Virus neutralization (VN) detects antibodies that neutralize infectious particles and prevent infection [8]. VN can be used to determine the magnitude of CPV antibody titres but it can only be performed in specialized laboratories [8].”

Line 51: “…has been evaluated by…”

Line 53: “…has been evaluated by…”

Line 56: “The aim of this study was to compare the four commercially available POC tests detecting antibodies against CPV concerning their practicability, specificity, sensitivity, positive predictive value (PPV), negative predictive value (NPV), as well as overall accuracy (OA) using VN as the reference standard.“

Line 87: “…overnight…”

Line 89: “...from a privately-owned vaccinated…“

Line 98: “The ImmunoComb® and the TiterCHEK® are enzyme-linked immunosorbent assays”

Line 123: “Sensitivity, specificity, PPV, NPV, and OA of…”

Line 125: “…were the tests with the highest specificities (83% and 67%, respectively)…”

Line 129:”… were the tests with the highest sensitivities (95% and 80%, respectively)…”

Line 135: “Statistical analysis revealed poor agreement between TiterCHEK® and FASTest®: kappa=0.124, FASTest® and ImmunoComb®: kappa=0.052, TiterCHEK® and ImmunoComb®: kappa=0.162, CanTiCheck® and ImmunoComb®: kappa=0.285, CanTiCheck® and FASTest®: kappa=0.052 (Table 6). There was a minimal agreement between CanTiCheck® and TiterCHEK®: kappa=0.344.”

Line 140: Table 2. Results of canine parvovirus antibody testing in sera of 198 dogs and comparison of the four point-of-care tests using VN as the reference standard

Line 152: “Table 3. Performance parameters of the four point-of-care tests to detect antibodies against canine parvovirus based on results given in table 1: sensitivity, specificity, positive predictive value, and negative predictive value,…”

Line 161: “overall accuracy, probability that a dog was correctly classified by the tests”

Line 184: “Table 6. Agreement of the results of the four point-of-care tests using Cohen´s kappa statistic”

Line 190: “Although disease caused by CPV infection is rare in Northern European countries today, the risk of spread of CPV via import of dogs from Southern or Eastern Europe is still high [17,18].”

Line 203: “both tests were unproblematic to perform.”

Line 204: “…buffered-mixed…”

Line 214: “…realized…”

Line 237: “The FASTest® has been evaluated…”

Line 258:” Hemagglutination inhibition (HI)“

Line 264: ”…without hemagglutinating properties…”

Yes

Can be improved

Must be improved

Not applicable

Does the introduction provide sufficient background and include all relevant references?

(x)

( )

( )

( )

Is the research design appropriate?

(x)

( )

( )

( )

Are the methods adequately described?

(x)

( )

( )

( )

Are the results clearly presented?

(x)

( )

( )

( )

Are the conclusions supported by the results?

(x)

( )

( )

( )

Comments and Suggestions for Authors

The paper entitled "Comparison of four commercially available point-of-care tests to detect antibodies against canine parvovirus in dogs" is a very useful, well written, and easily readable study.

We thank the reviewer for the positive comments.

It will have a significant impact in practice.

Yes, we agree.

I only found two typos:

line 208: buffered instead of bufferd

We made the correction.

Line 204: “…buffered-mixed…”

line 218: realized instead of realzied.

We made the correction.

Line 214: “…realized…”

Round 2

Reviewer 2 Report

The revised manuscript, regrettably, suffers from the same major defect as the original one: the low number of VN-negative samples makes it impossible to determine assay specificities, and hence to compare the assays with each other.

The main purpose of assays like these, in the field, is to find the unprotected animal. And the main purpose of a manuscript like this would be to rank the four assays in terms of diagnostic performance, most importantly with regards to specificity. With the existing samples (low number of negatives) this aim remains unfulfilled.

With the specificity 95% confidence intervals 12-88%, 22-96%, 4-78% and 36-100% of the present version, it would be misleading to give exact estimates of 50%, 33%, 67% and 83% for specificity. The same holds for the negative predictive values (1-14%, 1-13%, 2-52% and 4-25% versus exact values of 5%, 5%, 18% and 12%). Thus I would remove all references to exact specificity values, as well as all recommendations of assays in terms of superiority in specificity. Overall, it would need to be emphasised in Abstract, Discussion and Conclusions that, based on these data, no relevant conclusions can be drawn on assay specificities, and no single test can be said to be superior.

Altogether, under these circumstances the article would provide little significant (or interesting) information: even if all of the tests were quite sensitive, the important information on test specificities would remain absent. The observed high prevalence of neutralizing antibodies is of some interest, but beyond the manuscript's main scope.